# Experimental Study on Measuring and Tracking Structural Displacement Based on Surveillance Video Image Analysis

**DOI:** 10.3390/s24020601

**Published:** 2024-01-17

**Authors:** Tongyuan Ni, Liuqi Wang, Xufeng Yin, Ziyang Cai, Yang Yang, Deyu Kong, Jintao Liu

**Affiliations:** 1College of Civil Engineering, Zhejiang University of Technology, Hangzhou 310023, China; 2112106013@zjut.edu.cn (L.W.); 211122060043@zjut.edu.cn (X.Y.); 221122060099@zjut.edu.cn (Z.C.); yangyang@zjut.edu.cn (Y.Y.); jtliu@zjut.edu.cn (J.L.); 2Key Laboratory of Civil Engineering Structures & Disaster Prevention and Mitigation Technology of Zhejiang Province, Hangzhou 310023, China

**Keywords:** health monitoring, structural displacement, tracking and monitoring, image processing technology, surveillance video images

## Abstract

The digital image method of monitoring structural displacement is receiving more attention today, especially in non-contact structure health monitoring. Some obvious advantages of this method, such as economy and convenience, were shown while it was used to monitor the deformation of the bridge structure during the service period. The image processing technology was used to extract structural deformation feature information from surveillance video images containing structural displacement in order to realize a new non-contact online monitoring method in this paper. The influence of different imaging distances and angles on the conversion coefficient (*η*) that converts the pixel coordinates to the actual displacement was first studied experimentally. Then, the measuring and tracking of bridge structural displacement based on surveillance video images was investigated by laboratory-scale experiments under idealized conditions. The results showed that the video imaging accuracy can be affected by changes in the relative position of the imaging device and measured structure, which is embodied in the change in *η* (actual size of individual pixel) on the structured image. The increase in distance between the measured structure and the monitoring equipment will have a significant effect on the change in the *η* value. The value of *η* varies linearly with the change in shooting distance. The value of *η* will be affected by the changes in shooting angle. The millimeter-level online monitoring of the structure displacement can be realized using images based on surveillance video images. The feasibility of measuring and tracking structural displacement based on surveillance video images was confirmed by a laboratory-scale experiment.

## 1. Introduction

Surveillance video images have been widely used in real- and full-time traffic flow monitoring of highways and urban traffic, as well as in intelligent buildings [1] and urban security monitoring [2], and the big data of traffic flow surveillance video images (BDTFSVI) formed during the traffic flow monitoring operation plays an extremely important role in national road network planning, intelligent navigation, and traffic management [3]. The structural deformation information of the bridge structures during the service period was implied in some of these surveillance digital video images of the BDTFSVI. Structural displacement is one of the important indicators to describe the health state of the bridge structures during the service period, and it is one of the most important contents of bridge health monitoring [4,5,6]. The evaluation of bridge structural performance changes and safety status can be achieved by monitoring, analyzing, and identifying the bridge structural displacement response of the target structure of bridges [7,8,9]. The traditional method of bridge inspection relies on workers going to the bridge and employing ladders, scaffolding, or lifters to reach critical parts of the bridge that are not easily accessible [10,11]. In recent years, a novel non-contact deformation measurement method to detect bridge structural deformations by using digital images has developed rapidly with the development of computer technology and image processing technology [12,13,14,15]. Compared with traditional monitoring systems based on measuring point sensors, such as accelerometers, robotic total stations, etc., the application of the image-based monitoring system to estimate the service state of bridge structures has advantages such as holography, convenience, and economy [16,17,18]. Access devices for digital images usually include digital cameras [19,20,21,22,23], UAV cameras [10,24,25,26], industrial cameras [27,28,29,30], mobile phones [31,32,33], etc. Many researchers [34,35,36] have achieved success in using image processing techniques to detect cracks in concrete surfaces. D. Dias-da-Costal proposed a method of monitoring crack expansion in the structure by using image displacement. While the actual displacement will be tracked by the correlation of the photogrammetric image with the original reference image, the structural displacement in the target image can be matched to a specific analysis phase [37]. Francesco Mugnai compared the precision of commercial optical sensors (DSLR Nikon D3200) with a state-of-the-art Image Assisted Total Station (IATS) Leica Nova MS60 and assessed their accuracy. The results showed that image-based sensors are expected to gradually develop their potential in structural monitoring [19]. A.Z. Hosseinzadeh investigated the modal identification of building structures using vision-based measurements from multiple interior surveillance cameras [38,39]. Under idealized circumstances, laboratory-scale tests were used to assess the proposed vision-based method. This was achieved by using image processing to track feature points across multiple video frames as the spatial position changed between the structures. However, it is rare to perform research using traffic flow surveillance video images and big data to monitor the deformation status of the bridge during the operation period.

Therefore, a new kind of non-contact monitoring method for bridge structure deformation could be realized in the future: the localization information change of the prefabricated circular target in the surveillance video is used to characterize the displacement change of the structural location feature point through surveillance video analysis and image processing techniques, so as to track and monitor the structural displacement [40]. The value of the video image information can be acquired from the BDTFSVI as a process in Figure 1, and the database of bridge structural health monitoring can be established. Furthermore, intelligent automatic monitoring for bridge structure deformation can be achieved, combined with AI technology, and long-term continuous monitoring of structural deformation can be achieved by surveillance video images. Therefore, this new non-contact monitoring method is very meaningful to the modal warning of the bridge operation, and the economic and social benefits of that are very significant.

In this study, a novel non-contact monitoring method is proposed, which focuses on tracking and monitoring the development of structural displacement of small- and medium-span bridges under loads during operation and maintenance so as to realize two-dimensional measurements of the displacement response of the structure. The abovementioned bridge structure displacements are small compared with the structural scale and the distance of the camera-to-structure. In order to realize this new non-contact online monitoring method, the influence of different imaging distances and angles on the conversion coefficient (*η*) that converts the pixel coordinates to the actual displacement was first experimentally studied. Then, the measuring and tracking of bridge structural displacement based on surveillance video images was investigated by laboratory-scale experiments under idealized conditions. Lastly, the feasibility of measuring and tracking structural displacement based on surveillance video images is experimentally confirmed in this paper.

## 2. Materials and Methods

### 2.1. Experiment of Surveillance Video Images Calibration

The basic relationship between the perspective and location of a surveillance camera is shown in Figure 2. The coordinate of point *p’* in the imaging plane system is (*x_c_*, *y_c_*). The point “*p*” in Figure 2 is a point of target in the space, and the point “O” is the optical center of the surveillance camera. The point “*p*′ (*x_c_*, *y_c_*)” in the imaging plane was an image of the point “*p*” when the light of the point “*p*” passes through the optical center at point “O”. In this paper, the influence law will be determined through experimental analysis based on some experiments. The literature [31,32] had shown that the measurement of accuracy would be influenced by individual surveillance cameras with different shooting distances and angles. So, the calibration of the surveillance camera is an essential requirement before tracking structural displacement with surveillance images. In this study, a gun-type surveillance system produced by Hikvision (Hangzhou, China) was adopted, the model of which is the DS-3T86FWDV2 with a prime lens. The best shooting range for manufacturer product introduction is 1 m to 4 m. The equipment performance parameters are shown in Table 1, and this kind of equipment is very common not only in China but also other countries. Considering the possibility of future engineering applications, ordinary general equipment was selected in this study, and there were no special requirements for the equipment. In order to find out the internal and external parameters of the surveillance camera, as well as the distortion parameters, and to obtain images that meet the experimental requirements, a calibration experiment of the surveillance lens was designed, as shown in Figure 3.

The calibration method used for the surveillance camera in this study was proposed by Zhang Zhengyou [41]. Before the calibration experiment with this method, the checkerboard pattern was selected as a calibrator with known dimensions. The first step of the experiment was to print a black and white checkerboard pattern and stick it flatly on a smooth portable surface. The checkerboard was composed of black and white cells, and each cell size is 25 mm × 25 mm. Then, several surveillance images were taken with the checkerboard pattern in different directions and at different distances by moving the folder board that was pasted with a checkerboard pattern within sight of a surveillance camera. Particularly, the checkerboard pattern image should be evenly distributed in each position within the visual range of the camera to improve the accuracy of the calibration. Furthermore, interference information such as railings and stripe patterns should be avoided as they affect checkerboard recognition. More than 3 images (15–20 images as the optimal number) were selected for the calibration of the surveillance camera, and the linear solution of the internal parameters of the surveillance camera was obtained.

### 2.2. Algorithm Experiment of Surveillance Video Images Calibration

Image distortion refers to the original image distortion caused by the lens manufacturing precision and the deviations of the assembly process [42]. Lens distortion is distributed along the lens radius, and the main manifestations are barrel distortion and pincushion distortion [43]. Those characteristics indicate the unnatural deformation or distortion of the image. The lens distortion of a surveillance camera can be divided into radial distortion and tangential distortion. In turn, radial distortion and tangential distortion will produce two apparent forms of distortion in the image, which can be named ‘pincushion distortion’ and ‘barrel distortion’ as shown in Figure 4. The distorted image distorts the actual spatial position relationship of the real points in the image, and the corrected image pixels can reflect the actual spatial position relationship of the point. After radial distortion and tangential distortion were corrected, the *barrel distortion and* the *pincushion distortion* of images disappeared. The radial distortion can be corrected through correction computation using Equation (1), and the tangential distortion can be corrected through correction computation using Equation (2) [44].
(1)$$\left\{\begin{array}{l} x_{dr}=x\left(1+\kappa_{1}r^{2}+\kappa_{2}r^{4}+\kappa_{3}r^{6}\right)\\ y_{dr}=y\left(1+\kappa_{1}r^{2}+\kappa_{2}r^{4}+\kappa_{3}r^{6}\right) \end{array}\;\mathrm{radial}\;\mathrm{distortion}\right.$$



(2)
$$\left\{\begin{array}{l} x_{dt}=x+\left[2\lambda_{1}y+\lambda_{2}\left(r^{2}+2x^{2}\right)\right]\\ y_{dt}=y+\left[2\lambda_{1}x+\lambda_{2}\left(r^{2}+2y^{2}\right)\right] \end{array}\;\mathrm{tangential}\;\mathrm{distortion}\right.$$



Therefore, the radial distortion and tangential distortion can be corrected totally by Equation (3). The schematic of the correction principle and correction effect comparison are shown in Figure 4.
(3)$$\left\{\begin{array}{l} x_{0}=x\left(1+\kappa_{1}r^{2}+\kappa_{2}r^{4}+\kappa_{3}r^{6}\right)+2\lambda_{1}xy+\lambda_{2}\left(r^{2}+2x^{2}\right)\\ y_{0}=y\left(1+\kappa_{1}r^{2}+\kappa_{2}r^{4}+\kappa_{3}r^{6}\right)+\lambda_{1}\left(r^{2}+2y^{2}\right)+2\lambda_{2}xy \end{array}\right.$$
where (*x_dr_*, *y_dr_*) is a coordinate after the correction of the radial distortion, (*x_dt_*, *y_dt_*) is a coordinate after the correction of the tangential distortion, and (*x*_0_, *y*_0_) is a coordinate after the correction of the radial distortion and the tangential distortion; $\kappa_{1}$, $\kappa_{2}$, $\kappa_{3}$, $\lambda_{1}$, $\lambda_{2}$ are five correction parameters.

### 2.3. Experiment to Measure and Track Bridge Structural Displacement by Surveillance Video Image

#### 2.3.1. Experimental Instrument Used for Measuring and Tracking Bridge Structural Displacement

The specific test instruments for the measuring and tracking of bridge structure displacement (vertical displacement of simple support beam middle point) experiment using surveillance camera images are shown in Figure 5, such as the laser rangefinder, level gauge, circular target, and surveillance camera. The laser rangefinder was used to determine the distance between the target and the surveillance camera. The level gauge was also used to control the imaging height of the surveillance camera, which was always consistent with the same height of the circular target. The diameter of the circular target was 60 mm (as a region of interest, abbreviated: ROI), and the circular target was pasted on the surface of the object to be measured. The center of the images would be in correspondence with the screen center of the surveillance camera. The surveillance camera was fixed to a tripod head and was always in a horizontal position. The laser rangefinder was horizontally attached to the adjusted surveillance camera by using a viscous superglue. The laser emitted by the laser rangefinder was always parallel to the optical axis of the surveillance camera. Thus, the linear distance between the surveillance equipment and the target could be measured by the laser rangefinder.

#### 2.3.2. Experimental Method of Measuring and Tracking Bridge Structural Displacement

In this study, a method to characterize structural displacement was designed based on the digital image processing of the surveillance camera images [45,46]. In order to validate this method, an experiment for measuring and tracking the structural deformation of a simple support beam of aluminum alloy metal undergoing displacement indoors with the surveillance video images was investigated, and the experiment is described as follows:

Based on the collected surveillance images and the dynamic structural deformation information, the value image database file is formed through logical screening. Firstly, in order to extract structural feature point information and calculate displacement, the pixel coordinate *O* (*x*_0_, *y*_0_) in the center of the initial image was calculated by fitting the target coordinates by ellipse. Then, the change in coordinate information is obtained by fitting the circular target of the measured position of the ellipse, and the displacement change of the characteristic point of the corresponding position of the characterized structure can be obtained by using the conversion coefficient *η* that converts the pixel coordinates to the actual displacement. By fitting the image of a circular target, not only coordinate information but also dimension information can be obtained. Given the actual size of the target, the conversion coefficient *η_h_* (vertical direction: *η_v_*) can be calculated. That is, the bullseye coordinates *O* (*x_t_*, y_t_) were calculated at time *t*. The structural feature point displacement can be represented by the product of the change in the pixel’s horizontal coordinate Δ*x* (vertical: Δy) in the center of the target figure with the corresponding direction shifts from the extremely horizontal direction. Combined with the digital image collection of a surveillance camera to monitor the vertical displacement of close-distance shooting, the feasibility of the method for structural displacement measurement and the reliability of measurement accuracy can be verified in this way.

The change in surveillance’s posture and the position of the camera will affect the imaging effect of the monitoring device [1]. That is, the distance between the surveillance camera and the monitored structure will affect the geometric size of a single pixel of imaging from the surveillance camera [27]. Similarly, the impact of surveillance camera shooting angle changes will affect the geometric size of a single pixel of imaging on the surveillance camera. The relationship law of the geometric size of a single pixel at different distances between the surveillance camera and the measured structure can be sought experimentally. These rules can be used to calculate the coordinate change of the structural displacement change at the monitoring point.

Thus, the experiments on the effects of posture and position changes of surveillance cameras were conducted first before the experiment of measuring and tracking the structural deformation of aluminum alloy metal simple support beam displacement.

The linear distance from the surveillance camera to the target was changed during monitoring, and the laser rangefinder was used to determine the exact position points. The test station point was set up when the distance from surveillance equipment to target was from 0.5 m to 5.0 m and every 0.5 m apart, respectively. Then, the imaging pixel value of the measured circular target at each distance can be obtained through the monitoring images taken at different distances, and the parameter η, size of the calibrated single pixel point, can be calculated by combining with the actual size of the target.

In the shooting angle change experiment, the image device must be kept at the same height as the target and face the target squarely at the initial moment. After that, slowly rotate the surveillance camera by manipulating the bracket to achieve a certain angle between the center of the surveillance equipment screen and the target when the angle range of surveillance camera rotation is between 0° and 35°. In this way, a few surveillance sequence images containing circular targets can be obtained. The actual diameter of a circular target divided by the number of pixels it occupies in the image is the actual size of a single pixel dot, where the diameter of the fitted circle is given by the ellipse fitting function in the programming software.

#### 2.3.3. Environmental Conditions of Measuring and Tracking Experiment

In order to verify the feasibility of the method for structural displacement measurement and the reliability of measurement accuracy, a control measuring device was provided in this study. A high-precision laser displacement meter (Panasonic HGC1030, Produced in Shenzhen, China, the repeatability precision 0.01 mm, which is shown in Figure 5) was installed at the midpoint of the beam for controlling measurements. The vertical displacement changes at the beam midpoint were measured continuously and synchronously by using contact measurements. Obviously, the effects of environmental conditions on photogrammetry exist, such as the effects of light conditions, raining, or fog weather [47]. In this paper, the study was carried out under the determination of the light conditions in the laboratory room, and the influence of rain and fog meteorological conditions were not involved during the study process. The measured illumination in the laboratory of this study ranged from 50 to 20,000 lux.

## 3. Experimental Results

### 3.1. Calibrate Analysis on the Performance of Surveillance Camera

Obviously, the conversion coefficient *η* would be affected when the distance from the surveillance camera to the target was changed. These influences were revealed by experiments in this paper and the results are shown in Figure 6. The relative relationship between *η* (the conversion coefficient) and *d* (the monitoring distance) is presented as linear, and this law is similar to the literature [32]. The approximate expression of the curve was obtained by fitting with the function relation and this function can be expressed as Equation (4):(4)η=0.43677d−0.00081
where the *d* is the shooting distance of the surveillance camera. This linear rule is very helpful for the subsequent displacement calculation. Considering the effect of experimental error, and the effective number of coefficients, the Equation (4) can be modified to Equation (5).
(5)η=0.43677d

Combined with the results in the literature [32], the conversion coefficient *η* closely follows a linear relationship with the shooting distance *d*, and the slope of that is determined by the hardware performance.

### 3.2. The Influence of the Surveillance Device Posture on Calibration

Similar to distance changes, the parameter *η* would be also affected by imaging angle changes of the surveillance camera. The results of the relative relationship between the *η* values with different imaging angles at various special distances are shown in Figure 7. Due to the dispersion of the experimental data, the quantitative relationship between the actual size *η* of individual pixel dots and the imaging angle is difficult to visualize accurately. In order to obtain a functional relation of the experimental data and illustrate the mathematical and physical meanings behind the data, a single exponential decay function was used to fit the measured data. The original and substituted function could be expressed as Equations (6) and (7), respectively. Thus, the parameter *η* could be determined in different imaging angles and different distances through these functions.
(6)$$y=y_{0}+Ae^{-x/t}$$
(7)$$\eta =\eta_{0}+Ae^{-\theta /t}$$
where the *A* and *t* are two regression constants and *θ* is the imaging angles at different distances.

The results of Figure 7 showed that the overall trends of the fitted curves were highly in accordance with the measured curves. They demonstrated that a fitting curve can effectively reflect the relative relationship between the value of *η* and the imaging angle of a surveillance camera. The value of *η* tended to monotonically increase with the increase in the imaging angle of the surveillance camera and showed exponential growth in the functional relationship. This rule would be used in the subsequent displacement calculation too.

The specific parameters of the fitting curve of the quantitative relation between the *η* value and the surveillance camera’s imaging angle at each distance are shown in Table 2. The linear regression coefficient *R*^2^ values of each fitting curve were greater than 0.99, which indicated that the fitting function was highly in agreement with the experimental data. This also represented that the fitting function curve can accurately reflect the actual relationship between different structure positions and *η* values. The values of the index parameter *t*_1_ of all fitted curves were similar, and they ranged from −16 to −20. Therefore, the basic trend of *η* value changing with imaging angle at different distances was similar. It is important to note that the best applicable range of the surveillance camera selected for this experiment was from 1 to 4 m; the measurement accuracy of the surveillance camera decreased after more than 4 m. The actual application needed to determine the type of surveillance camera according to demand.

### 3.3. The Influences of Differing Distances and Horizontal Angles on the Conversion Coefficient η

In order to investigate the effect of structural position change on the accuracy of structural displacement tracking and monitoring, the variations in *η* with imaging angle at different distances were integrated. The results in Figure 8 show the *η* variations of the surveillance camera at different positions. The trend of *η* with angle is essentially unchanged. The extension of the linear distance between the target and surveillance camera and the increase in the imaging angle leads to the growth of the *η* value.

For the sake of revealing more intuitively the relationship of the *η* value with the relative position between the surveillance camera and the measured structure, the three-dimensional model of *η* with respect to imaging distance and imaging angle was plotted and it is shown in Figure 9.

However, the simple 3D model building did not concretely describe the quantitative relationship between *η* and the relative position of the structure. A function to describe this relationship can be established. Thus, the relevant module of the MATLAB R2021b software was used to establish coupling analysis of *η*, imaging distance, and imaging angle in order to improve the practical application value of experiment results. The second order polynomial function was applied to the three-dimensional fitting, since there were two independent variables. The original function can be expressed as Equation (8):(8)$$f(x,y)=p_{00}+p_{10}x+p_{01}y+p_{20}x^{2}+p_{11}xy+p_{02}y^{2}$$

The relationship equation obtained from the three-dimensional fitting is shown by Equation (9):(9)$$\eta (d,\theta )=-0.06164+0.44360d-0.00261\theta -0.00309d^{2}+0.00092d\theta +0.00008\theta^{2}$$

The intuitive image of the relative relationship is given in Figure 10. Based on these, *η* could be calculated if the imaging angle and imaging distance were obtained so that the structural displacement could be measured by the of the value of *η* and the magnitude of the displacement presented in the image. It should be noted that the formulas above for *η* are only applicable to the surveillance cameras selected in this paper. The relative relationship between *η* and the position of the structure should be obtained through experiment again if the lens or camera is replaced. Thus, the law of conversion coefficient *η* can accurately reflect the influences of posture and position changes of the surveillance camera.

## 4. Experimental Confirmations about the Feasibility of Measuring and Tracking Structural Displacement Based on Surveillance Video Images

The vertical deflection change of the center of the aluminum alloy beam under different loads was monitored by surveillance images. The different vertical static loads were simulated with different static weights, and the moving loads were simulated with a certain moving weight. Then, the displacement of the aluminum alloy metal guide rail beam structure was tracked and captured by means of the surveillance camera and self-developed program. The stability of the surveillance camera was less disturbed by environmental factors (such as breeze) during the measurement process because it was mounted on a fixed tripod to simulate its practical use.

### 4.1. Time Registration for Tracking and Monitoring

The high-resolution surveillance camera captured a high-quality digital image. However, the transmission process still takes time even if high-speed optical fiber transmission is adopted due to the large information capacity [48]. On the other hand, both image processing and displacement calculations will take some time. Therefore, there is a time delay Δ*t* between the data and the corresponding displacement deformation (as shown in Figure 11a) when calculating the displacement information. Therefore, the influence of Δ*t* should be eliminated to ensure the accuracy of tracking and monitoring to reflect structural deformation. In the process of structural displacement monitoring, the displacement curves measured by the laser displacement meter and the digital monitoring system have time errors, and temporal registration is adopted by data processing. After that, the effect of this time lag on monitoring can be eliminated, as shown in Figure 11b. The timing sequence of data from multiple data sources can be unified by temporal registration. Base on this, more monitoring equipment and monitoring big data can access or join this open measurement system in the future.

### 4.2. Video Images Tracking and Monitoring Structural Displacements

For construction management, maintenance, and management of the engineering structure, real-time monitoring is an important technology and key process [40,49]. In this study, the tracking and monitoring experiment was conducted in two parts. Firstly, a circular target was pasted in the middle position of the aluminum alloy, simply supported by the beam, and the sequence images were collected by the surveillance camera at a determined distance from the target (the distance of the experiment was 2 m). Self-developed software was run on the laptop to track and monitor the displacement of the target. Then, the load-holding state of the structure would be changed. Three equivalent-mass weights were successively loaded at the same position on the left side of the simply supported beam at the 4th second, 11th second, and 18th second. The weights were left for a period of time, and the weights were unloaded at 25th second, 33rd second, and 41st second, respectively. In the second test, the duration of the load increased. The times of load application were set at 51 s, 60 s, and 70 s, respectively, and the weights were unloaded at 79 s, 88 s, and 97 s, in the same order. The above experiment was repeated to avoid the experimental contingency. The displacement measured by the surveillance camera and displacement meter after temporal registration are shown in Figure 12.

Similarly, the tracking and monitoring experiment was also conducted under moving loads. A cylindrical weight was vertically erected on the outer position of the aluminum alloy beam, and its curved surface adhered to the surface of the aluminum alloy beam. Then, the weight was made to move to the center at a uniform speed and return to the starting position the same way. The displacement of the aluminum alloy beam with load movement obtained by the displacement meter measurement and a surveillance camera is shown in Figure 13.

The results of Figure 13 illustrate that there were still some problems with the temporal registration during the results comparison of the two different measurement methods. The current study could not find the precise cause at this time. It may be due to the insufficiency of the monitoring video bit rate or a delay in image information transmission for limited hardware facilities. Therefore, secondary manual correction was required. The comparison image after secondary temporal registration is shown in Figure 14.

### 4.3. Experimental Analysis for Measuring and Tracking Structural Displacement Based on Surveillance Video Images

The measurements from a high-accuracy laser displacement meter will be used as a benchmark to verify the accuracy of the surveillance camera in tracking and monitoring the displacement of the structure. The absolute error of the structural displacement monitoring experiment is shown in Figure 15. The results of structural displacement measurements using surveillance cameras showed a high degree of accuracy. Compared with reference measurements, the error rate of surveillance camera images with a deformation less than 0.15 mm was 6.13%; with a deformation less than 0.2 mm, it was 3.21%; with a deformation less than 0.25 mm, it was 1.63%; and with a deformation less than 0.3 mm, it was 0.44%. Furthermore, the monitoring error from the static load section was significantly smaller than the error from the moving load section. The error of the virtual circle center extraction increased as a result of the aluminum beam’s small vibrations as the load moved. In the static load section, the error rate with a deformation less than 0.2 mm would be up to 0.44%. So, the experiment results show that the high feasibility and accuracy of using surveillance cameras for structural displacement monitoring are acceptable.

In general, the relative error is more reflective than the absolute error in terms of the degree of confidence in the measurement. In this paper, the measured value of a high-precision displacement meter will be used as the conventional true value to verify the reliability of surveillance camera tracking and monitoring of structural displacement. The relative error of the structural displacement measured by using the surveillance camera is shown in Figure 16. Compared with reference measurement during the load-holding phase, the measurement data from surveillance camera images with a relative error lower than 15% arrived at 97.1%, and images with a relative error lower than 10% arrived at 95.6%. It is worth noting that the relative error curve shows a violent vibration at very few points. The reason was that the aluminum beam would oscillate and sway when the load changed suddenly. Overall, the calculation’s relative error value is low, and the trend is largely consistent.

## 5. An Outlook on New Full- and Real-Time Non-Contact Monitoring

The effects of environmental conditions (e.g., lighting conditions, atmospheric refraction, and turbulence) on the method of this paper would be further explored in our continuous studies to improve the quality of vision-based measurements. Full- and real-time non-contact monitoring of structural displacements will be realized in future practical applications. The approach used in this study may be further popularized and applied to the monitoring of bridge structure displacement, and additional monitoring tools may be connected to the measurement system to realize the further integration of monitoring big data, resulting in a new online monitoring technology for the entire life cycle of bridge service. Additionally, this approach can be used with the BDTFSI in the future to assess the deformation of the bridge using the distinctive details of the traffic flow image.

## 6. Conclusions

In this paper, the influence of different imaging distances and angles on the conversion coefficient (*η*) that converts the pixel coordinates to the actual movements was studied. Based on surveillance video image analysis and digital image processing, a method to characterize the displacement of structural position feature points was designed and a new non-contact measurement method of structural dis-placement was proposed. The measurement results of surveillance video images were compared with a high-precision laser displacement meter. The main conclusions are as follows:(1)The imaging accuracy can be affected by changes in the relative position of the imaging device and measured structure, which is embodied in the change in *η* (the actual size of an individual pixel) on the structured image.(2)The increase in distance between the measured structure and the monitoring equipment will have a significant effect on the change in *η*, and the value of *η* varies linearly with the change in shooting distance.(3)The value of *η* will be affected by changes in imaging angle. With the increase in the shooting angle, the value of *η* keeps increasing, and the change degree presents an exponential function relationship.(4)A new non-contact measurement method using surveillance video images was proposed, and the feasibility of measuring and tracking structural displacement based on surveillance video images was experimentally confirmed.(5)The changes in coordinates of the circular target center obtained by the ellipse fitting method are used to characterize the displacement of the corresponding feature points of the structure, and millimeter-level online monitoring of the structure displacement can be realized based on surveillance video images. The absolute error of this method compared with the high-precision displacement meter is less than 0.15 mm, and the relative error is less than 10%.

## Figures and Tables

**Figure 1 sensors-24-00601-f001:**
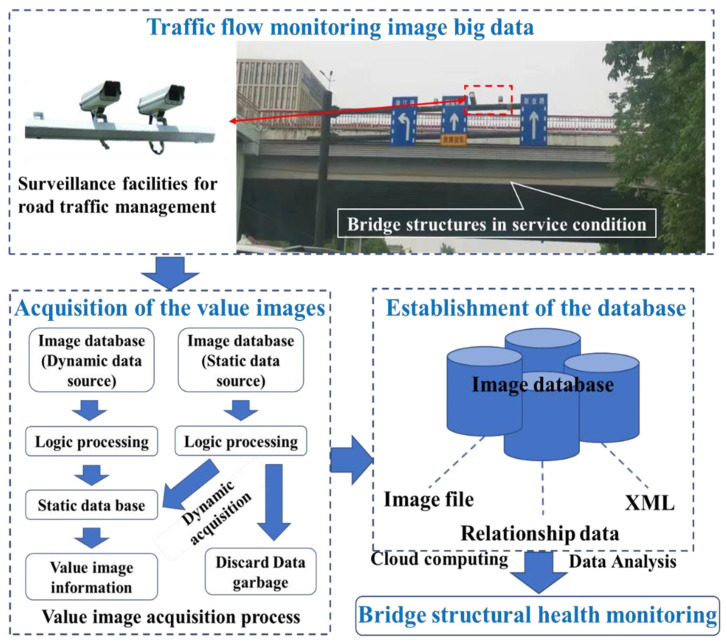
Road surveillance equipment and value video image information acquisition process.

**Figure 2 sensors-24-00601-f002:**
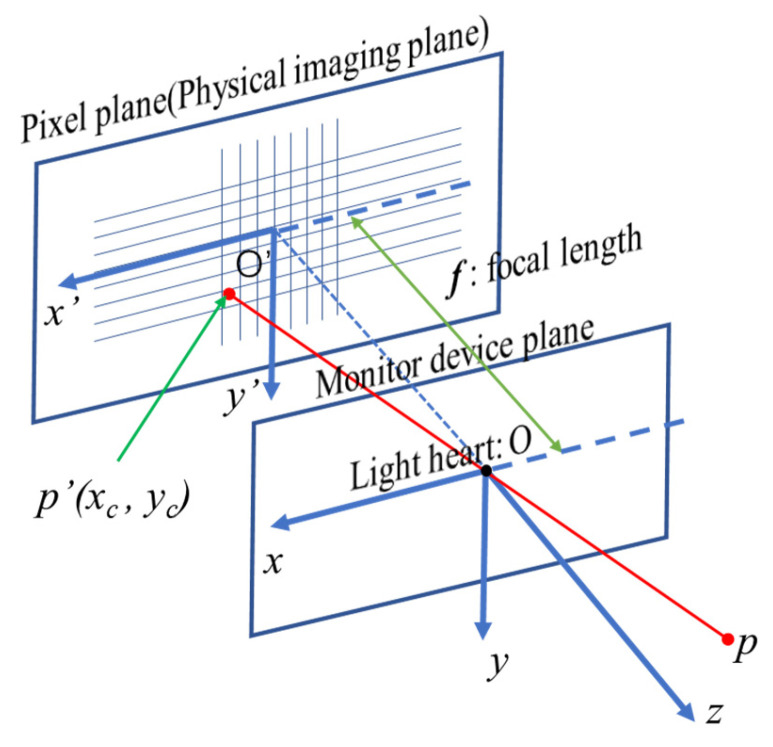
The relationship between perspective and location.

**Figure 3 sensors-24-00601-f003:**
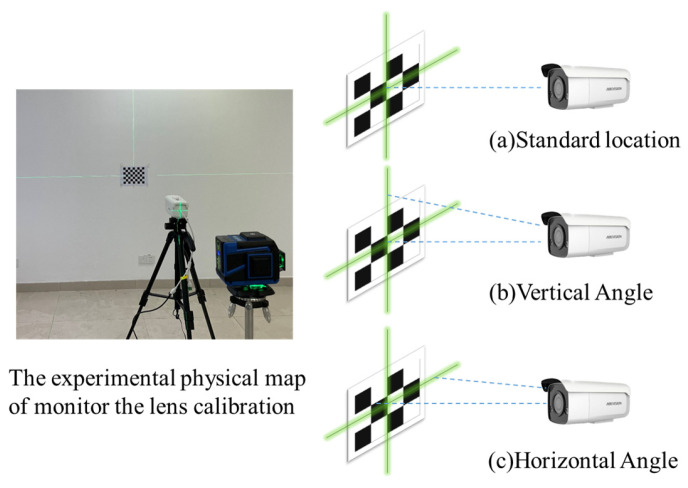
The calibration experiment using a checkerboard pattern.

**Figure 4 sensors-24-00601-f004:**
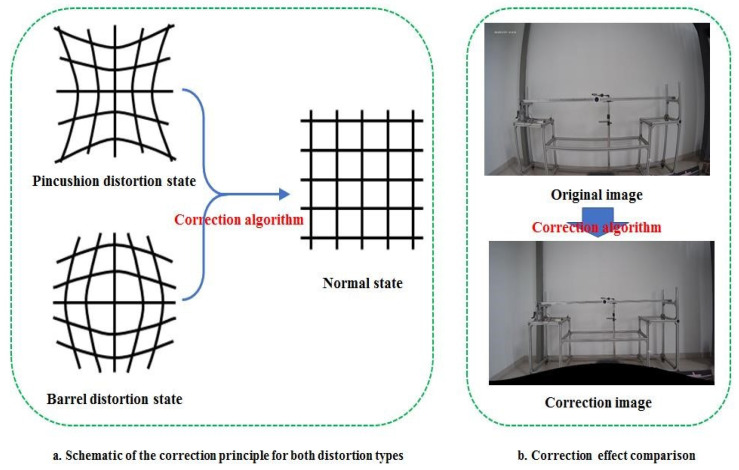
Schematic of the correction principle and correction effect comparison.

**Figure 5 sensors-24-00601-f005:**
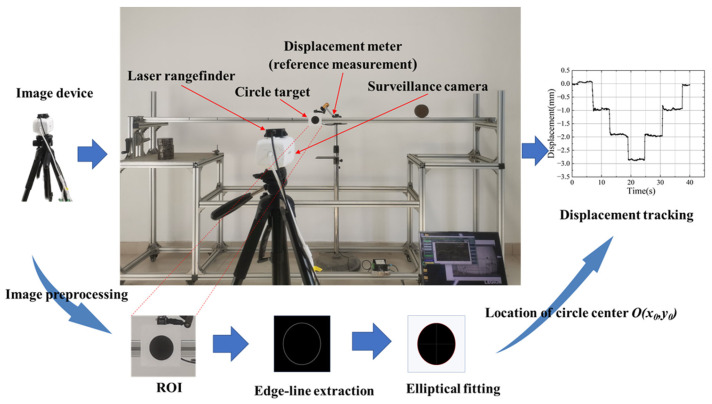
Structural displacement measurement experiment scheme using surveillance cameras.

**Figure 6 sensors-24-00601-f006:**
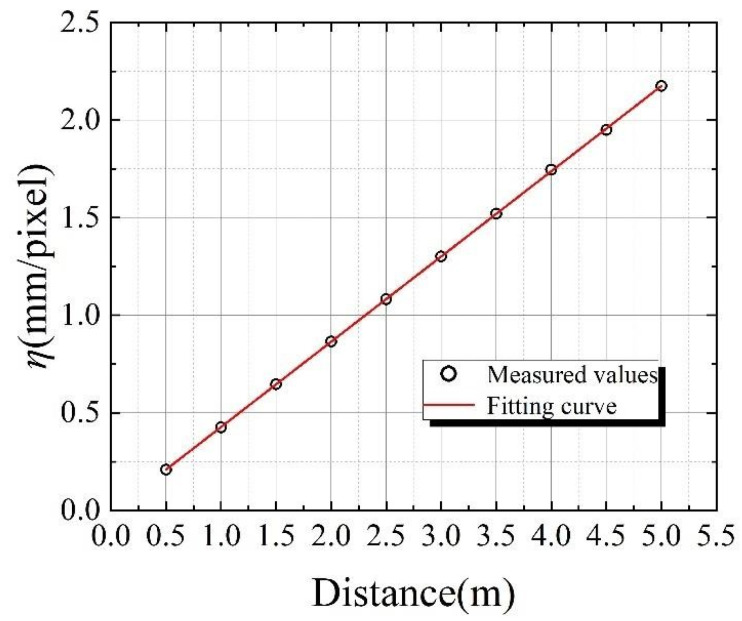
The relationship between *η* and ***d***.

**Figure 7 sensors-24-00601-f007:**
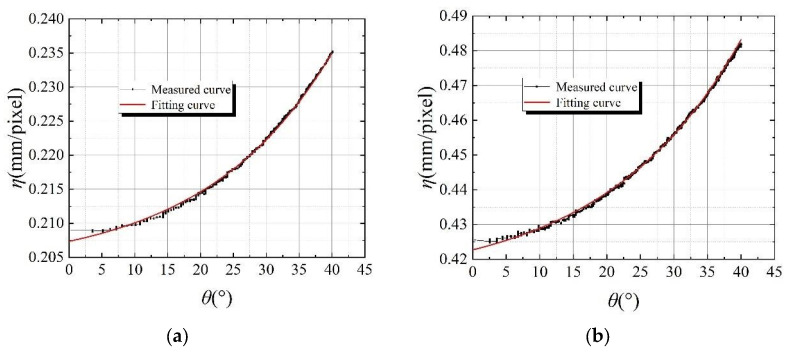
The relationship between *η* and ***θ***. (**a**) a distance of 0.5 m; (**b**) a distance of 1.0 m; (**c**) a distance of 1.5 m; (**d**) a distance of 2.0 m; (**e**) a distance of 2.5; (**f**) a distance of 3.0 m; (**g**) a distance of 3.5 m; (**h**) a distance of 4.0 m; (**i**) a distance of 4.5 m; and (**j**) a distance of 5.0 m.

**Figure 8 sensors-24-00601-f008:**
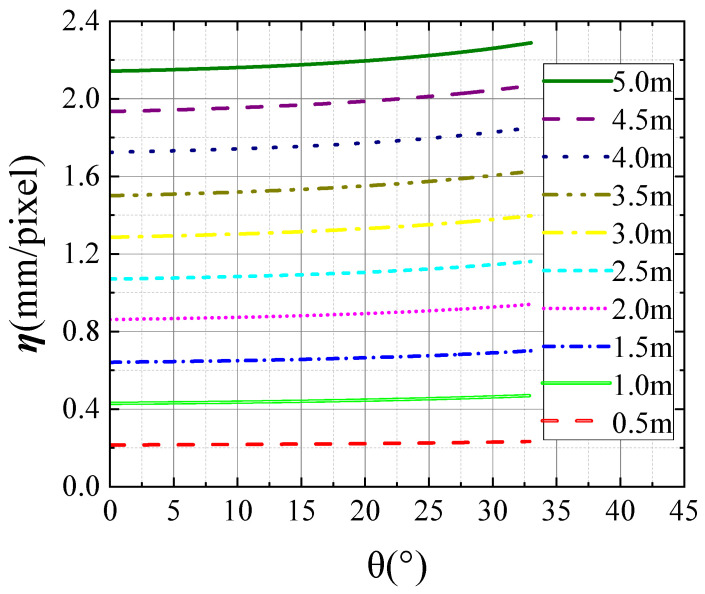
The relative relationship between *η* and structure position.

**Figure 9 sensors-24-00601-f009:**
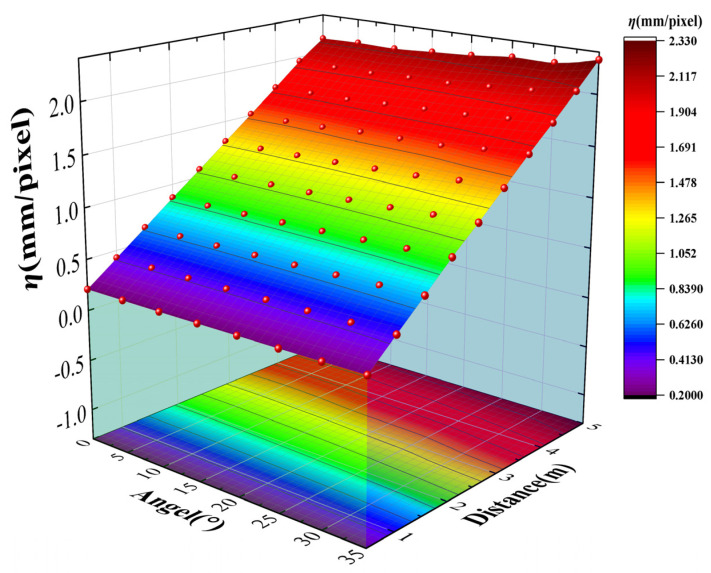
The relationship between the *η* value and imaging posture.

**Figure 10 sensors-24-00601-f010:**
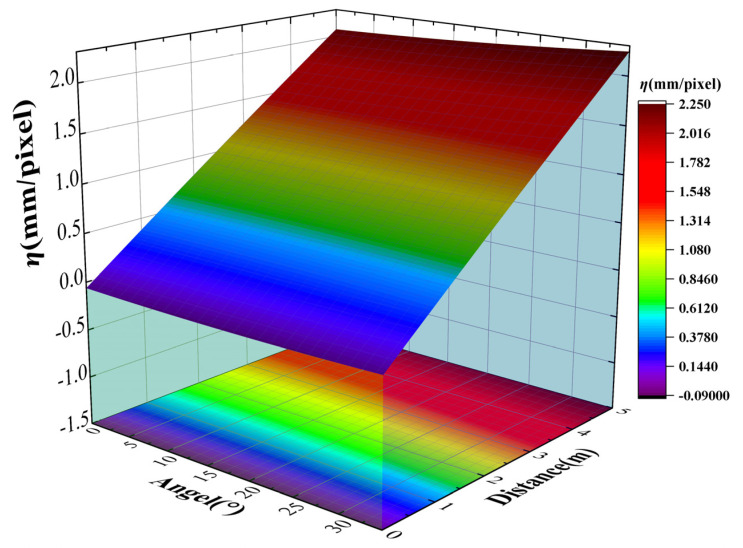
Three-dimensional fitting between *η* and the location of the structure (***d***, ***θ***).

**Figure 11 sensors-24-00601-f011:**
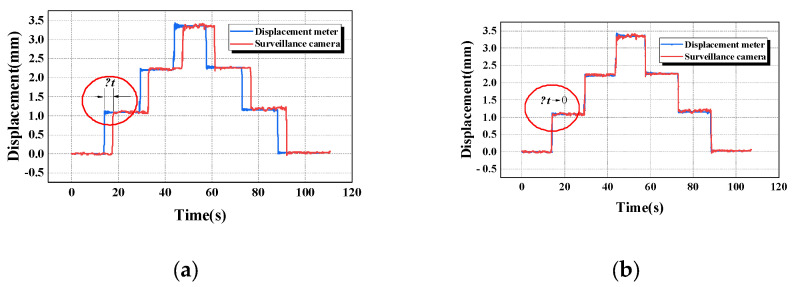
Time registration for tracking and monitoring (**a**) without eliminating time error; (**b**) eliminating time error.

**Figure 12 sensors-24-00601-f012:**
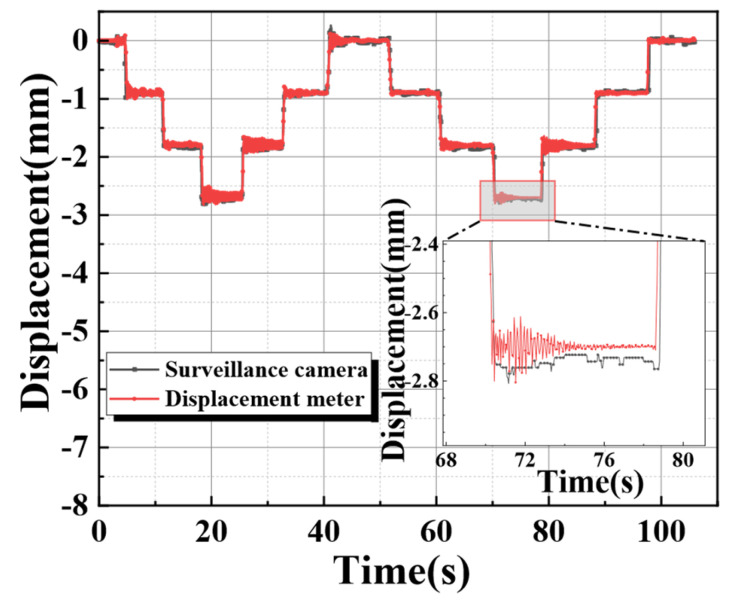
Displacement meter monitoring and digital image tracking and monitoring under static load.

**Figure 13 sensors-24-00601-f013:**
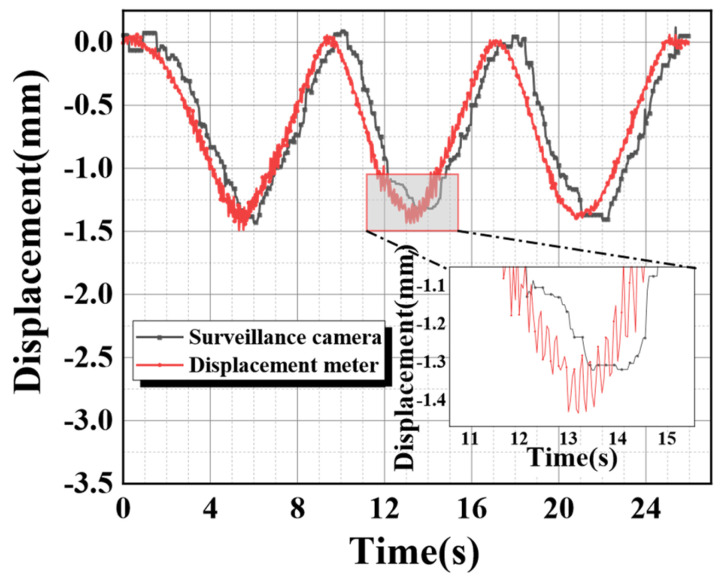
Displacement meter monitoring and image tracking and monitoring under moving load.

**Figure 14 sensors-24-00601-f014:**
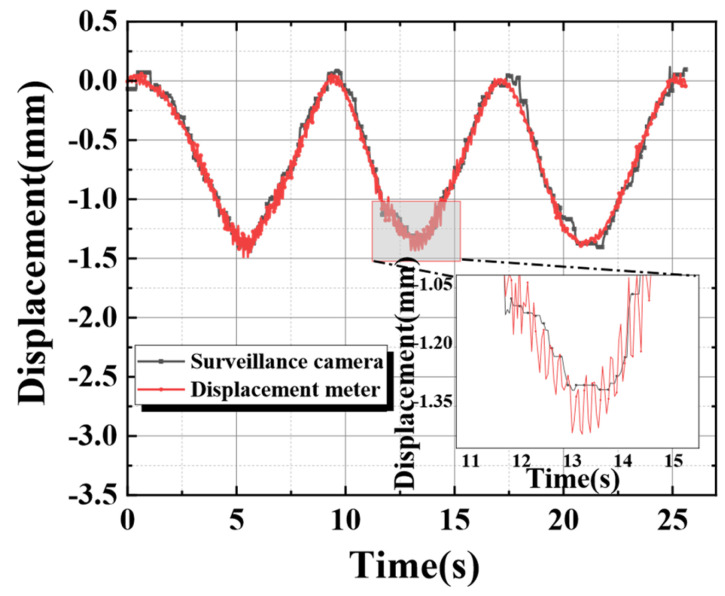
Tracking and monitoring after the secondary time registration.

**Figure 15 sensors-24-00601-f015:**
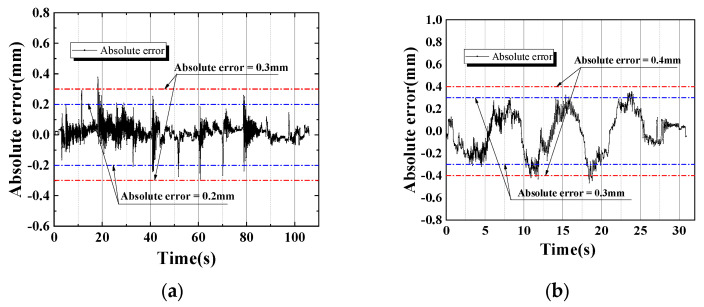
Absolute error of structural deformation monitoring experiments with (**a**) static load; (**b**) moving load.

**Figure 16 sensors-24-00601-f016:**
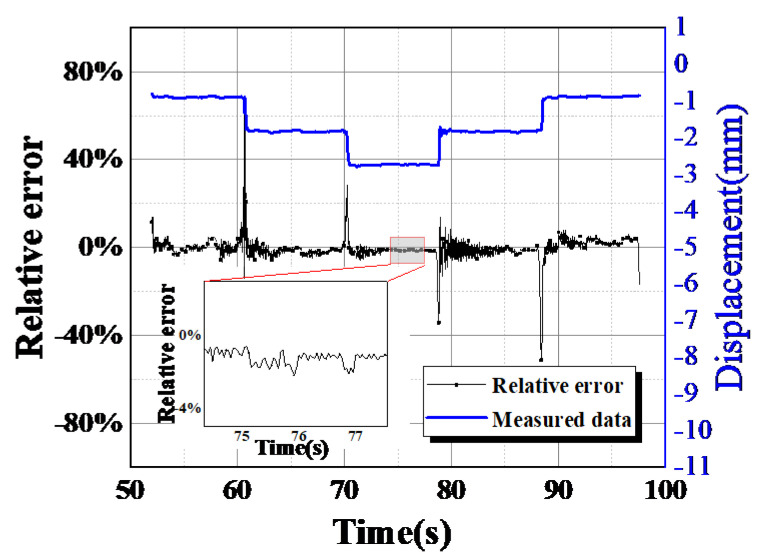
Relative error of structural deformation monitoring experiments.

**Table 1 sensors-24-00601-t001:** Equipment performance parameters of the surveillance camera.

Sensor Type	Resolution Ratio /Pixel	Focal Length /mm	The Frame Rate /Hz	Illumination /Lux
1/1.8″ Progressive Scan CMOS	3840 × 2160	2.8	25	0.009

**Table 2 sensors-24-00601-t002:** Fitting function parameters.

**Distance**	**0.5 m**	**1.0 m**	**1.5 m**	**2.0 m**	**2.5 m**
intercept *y*_0_	0.203	0.413	0.631	0.846	1.055
*t* _1_	−19.399	−20.339	−17.749	−18.431	−17.528
*R* ^2^	0.997	0.998	0.998	0.998	0.998
**Distance**	**3.0 m**	**3.5 m**	**4.0 m**	**4.5 m**	**5.0 m**
intercept *y*_0_	1.262	1.474	1.703	1.907	2.121
*t* _1_	−19.221	−18.913	−17.216	−18.483	−16.017
*R* ^2^	0.999	0.999	0.998	0.998	0.997

## Data Availability

Data are contained within the article.

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
