# Peer review of "Experimental Study on Measuring and Tracking Structural Displacement Based on Surveillance Video Image Analysis"

_sensors, 2024, doi:10.3390/s24020601_

Round 1

Reviewer 1 Report

Comments and Suggestions for Authors

The paper ‘Experimental study on measuring and tracking structural displacement base on surveillance video images’,

By Tongyuan Ni et al,

explores a novel approach to structural displacement monitoring using video frames.

More specifically, the Authors focus on monitoring the deformation of bridge structures during their service life using surveillance video images and image processing technology, which is indeed a clever concept, which addresses a specific gap in the field of bridge monitoring.

Thus, the main question of the paper is to prove the feasibility of this approach

Overall, the paper is well-structured and detailed. The article fits very well with the aims of the Journal and the Special Issue ‘AI Enhanced Civil Infrastructure Safety’; it also represents a relevant addition to the field in comparison with the already-published material.

Nevertheless, Several issues, related to both the paper’s content and format, are enlisted here below and should be addressed in order to achieve full acceptance.

1.      The Methodology section outlines the methods and techniques employed in the research, including the experimental setup, the types of cameras used, the parameters considered for structural displacement measurements, as well as the image processing algorithms used for tracking and measuring displacements. However, it is not totally clear if the laboratory instrumentation used to this purpose is directly comparable to the hardware cameras most commonly deployed in urban areas (not only in China but also in other countries)

2.      Eq. 4: it is not clear how general this linear rule is, i.e., if it can be applied also to other pieces of hardware in different contexts.

3.      Table 2 (and elsewhere): please use the same number of decimal digits (e.g. 1.055 instead of 1.0545)

4.      The authors correctly highlight the advantages of their proposed approach, emphasizing its economic and convenient nature. To this aim, they also report several recent studies on the broader topic of non-contact, video-based SHM. However, the state-of-the-art can be further expanded with related works such as https://doi.org/10.1111/str.12336, https://doi.org/10.1155/2021/5518163, https://doi.org/10.1117/12.2257831, and https://doi.org/10.1016/j.measurement.2017.02.024.

5.      Figure 1 is a bit too complicated and could be improved e.g. with a more straightforward flowchart.

6.      The resolution of some Figures (e.g. Fig. 2 and 3) is  low, please provide a better version of these pictures

7.      Figure 5 has many writings on it, which make it difficult to read. Also, many writings are superimposed on arrows and other graphical objects, again making it a bit difficult to appreciate them.

8.      Having Figures 9 and 10 more visually similar would be better.

9.      The Title of Section 3, is too long. It could be just reduced to “Experimental Results”.

10.   the conclusions are consistent with the evidence and arguments presented and address the main question posed. However, they could briefly recall also some numerical values from the Experimental results section

 The paper ‘Experimental study on measuring and tracking structural displacement base on surveillance video images’,

By Tongyuan Ni et al,

explores a novel approach to structural displacement monitoring using video frames.

More specifically, the Authors focus on monitoring the deformation of bridge structures during their service life using surveillance video images and image processing technology, which is indeed a clever concept, which addresses a specific gap in the field of bridge monitoring.

Thus, the main question of the paper is to prove the feasibility of this approach

Overall, the paper is well-structured and detailed. The article fits very well with the aims of the Journal and the Special Issue ‘AI Enhanced Civil Infrastructure Safety’; it also represents a relevant addition to the field in comparison with the already-published material.

Nevertheless, Several issues, related to both the paper’s content and format, are enlisted here below and should be addressed in order to achieve full acceptance.

1.      The Methodology section outlines the methods and techniques employed in the research, including the experimental setup, the types of cameras used, the parameters considered for structural displacement measurements, as well as the image processing algorithms used for tracking and measuring displacements. However, it is not totally clear if the laboratory instrumentation used to this purpose is directly comparable to the hardware cameras most commonly deployed in urban areas (not only in China but also in other countries)

2.      Eq. 4: it is not clear how general this linear rule is, i.e., if it can be applied also to other pieces of hardware in different contexts.

3.      Table 2 (and elsewhere): please use the same number of decimal digits (e.g. 1.055 instead of 1.0545)

4.      The authors correctly highlight the advantages of their proposed approach, emphasizing its economic and convenient nature. To this aim, they also report several recent studies on the broader topic of non-contact, video-based SHM. However, the state-of-the-art can be further expanded with related works such as https://doi.org/10.1111/str.12336, https://doi.org/10.1155/2021/5518163, https://doi.org/10.1117/12.2257831, and https://doi.org/10.1016/j.measurement.2017.02.024.

5.      Figure 1 is a bit too complicated and could be improved e.g. with a more straightforward flowchart.

6.      The resolution of some Figures (e.g. Fig. 2 and 3) is  low, please provide a better version of these pictures

7.      Figure 5 has many writings on it, which make it difficult to read. Also, many writings are superimposed on arrows and other graphical objects, again making it a bit difficult to appreciate them.

8.      Having Figures 9 and 10 more visually similar would be better.

9.      The Title of Section 3, is too long. It could be just reduced to “Experimental Results”.

10.   the conclusions are consistent with the evidence and arguments presented and address the main question posed. However, they could briefly recall also some numerical values from the Experimental results section

Comments on the Quality of English Language

Overall, the quality of English is good.

Author Response

Response to Reviewer 1 Comments

Point 1: The Methodology section outlines the methods and techniques employed in the research, including the experimental setup, the types of cameras used, the parameters considered for structural displacement measurements, as well as the image processing algorithms used for tracking and measuring displacements. However, it is not totally clear if the laboratory instrumentation used to this purpose is directly comparable to the hardware cameras most commonly deployed in urban areas (not only in China but also in other countries)

Response 1: Thank you very much for the comment. The resolution ratio is a performance index n to reflect the level of technology and market application breadth of surveillance camera, and the resolution ratio of surveillance camera which we used in this paper is 3840×2160 pixel (It's about 8 megapixels), and this kind of equipment is very common not only in China but also abroad. Considering the possibility of future engineering application, ordinary general equipment was selected in this study, and there are no special requirements for the equipment. The performance indicators have been explained in the paper, and supplementary explanations are made in order to express them more clearly such as follows:

Line 106:

“The equipment performance parameters are shown in Table 1, and this kind of equipment is very common not only in China but also abroad. Considering the possibility of future engineering application, ordinary general equipment was selected in this study, and there are no special requirements for the equipment.”

Point 2:   Eq. 4: it is not clear how general this linear rule is, i.e., if it can be applied also to other pieces of hardware in different contexts.

Response 2: Thank you very much for the comment. The meaning of the paper was not expressed clearly, and modification has been made as follows:

Line 247

“These influences were revealed by experiments in this paper and the results were shown in Figure 6. The relative relationship between η (the conversion coefficient) with d (the monitoring distance) presented as linear, and this law was similar to the literature [32]. The approximate expression of the curve was obtained by fitting with the function relation and this function can be expressed as Equation (4):

                           (4)

Where the d is the shooting distance of surveillance camera. This linear rule is very helpful for the subsequent displacement calculation. Considering the effect of experimental error, the Equation 4 can be modified to Equation (4').

                                    (4’)

Combined with the results of literature [32], the conversion coefficient η follows a linear relationship with the shooting distance d well, and the slope of that is determined by the hardware performance.

Point 3: Table 2 (and elsewhere): please use the same number of decimal digits (e.g. 1.055 instead of 1.0545)

Response 3: Thank you very much for the comment. We modified it.

Table 2. Fitting function parameters

Distance

0.5m

1.0m

1.5m

2.0m

2.5m

intercept y0

0.203

0.413

0.631

0.846

1.055

t1

-19.399

-20.339

-17.749

-18.431

-17.528

R2

0.997

0.998

0.998

0.998

0.998

Distance

3.0m

3.5m

4.0m

4.5m

5.0m

intercept y0

1.262

1.474

1.703

1.907

2.121

t1

-19.221

-18.913

-17.216

-18.483

-16.017

R2

0.999

0.999

0.998

0.998

0.997

Point 4: The authors correctly highlight the advantages of their proposed approach, emphasizing its economic and convenient nature. To this aim, they also report several recent studies on the broader topic of non-contact, video-based SHM. However, the state-of-the-art can be further expanded with related works such as https://doi.org/10.1111/str.12336, https://doi.org/10.1155/2021/5518163, https://doi.org/10.1117/12.2257831, and https://doi.org/10.1016/j.measurement.2017.02.024.

Response 4: Thank you very much for the comment. We have learned the relevant literatures, and we have benefited a lot.

Point 5: Figure 1 is a bit too complicated and could be improved e.g. with a more straightforward flowchart.

Response 5: We gratefully appreciate for your valuable comment. The figure 1 is not only a flowchart for this paper; it covers the purpose of research, the object of service and sources of research work’s images, and future direction of our research team. We think that a simplify flowchart would not express completely the significance of the study. Thanks again for your valuable comment.

Point 6: The resolution of some Figures (e.g. Fig. 2 and 3) is low, please provide a better version of these pictures.

Response 6: We are sorry for the inconvenience brought to the reviewer. The Figures 2 and 3 have been modified accordingly, and the modified version of these two pictures were provided as follows:

Figure 2. The relationship between perspective and location

Figure 3. The calibration experiment using checkerboard pattern

Point 7: Figure 5 has many writings on it, which make it difficult to read. Also, many writings are superimposed on arrows and other graphical objects, again making it a bit difficult to appreciate them.

Response 7: Thank you very much for the comment. The Figure 5 has been modified accordingly to reviewers' suggestions as follows:

Figure 5. Structural displacement measurement experiment scheme using surveillance cameras

Point 8: Having Figures 9 and 10 more visually similar would be better.

Response 8: We think we see about the reviewer’s concern. The Figure10 was modified as follows:

Figure 10. Three-dimensional fitting between η and the location of the structure (d, θ)

Point 9: The Title of Section 3, is too long. It could be just reduced to “Experimental Results”.

Response 9: We gratefully appreciate for your valuable suggestion. The title of section 3 has been

modified to “Experimental Results”.

Point 10: The conclusions are consistent with the evidence and arguments presented and address the main question posed. However, they could briefly recall also some numerical values from the Experimental results section.

Response 10: Thank you very much for the comment. We have modified our conclusions accordingly, as follows:

Line 475:

“(5) The changes in coordinates of the circular target center obtained by the ellipse fitting method are used to characterize the displacement of the corresponding feature points of the structure, and millimeter-level online monitoring of the structure dis-placement can be realized based on surveillance video images. The absolute error of this method compared with the high-precision displacement meter is basically less than 0.15mm, and the relative error is basically less than 10%.”

Reviewer 2 Report

Comments and Suggestions for Authors

The authors suggest that they propose a method for structural monitoring based on analysis of common surveillance video images.  In reality, they propose a method for calibration and sensitivity analysis of certain parameters of short-range positioning of a target from video analysis.  This is quite different. This means that the title and the focus of the manuscript should be changed.

Perhaps the proposed technique designed/tested for distances up to a few meters might be useful for structural monitoring, but this requires understanding of what should be measured: 3-D to 1-D coordinates, a range of expected displacements from millimeters to meters, and targets at a distance of a few tens of meters to a few kilometers, depending on the type and dimensions of monitored structure, with measurements under the influence of atmospheric effects.  Can the proposed technique respond to such needs?  I think that the authors should first consult two papers on this topic, Xu & Brownjohn, Civ. Struct. Health Monit. 2018, 8, 91–110 and Fradelis et al, Sensors 2020, 20, 3217, and then try to place their idea in this context.

If the authors respond positively to the above question, a radical revision of the manuscript is required, and this may lead to a manuscript which I shall be happy to recommend for publication in Sensors.

In addition, careful copy editing is required (in references, etc.)

Comments on the Quality of English Language

Minor copy editing required

Author Response

Response to Reviewer 2 Comments

Point 1: The authors suggest that they propose a method for structural monitoring based on analysis of common surveillance video images.  In reality, they propose a method for calibration and sensitivity analysis of certain parameters of short-range positioning of a target from video analysis.  This is quite different. This means that the title and the focus of the manuscript should be changed.

Perhaps the proposed technique designed/tested for distances up to a few meters might be useful for structural monitoring, but this requires understanding of what should be measured: 3-D to 1-D coordinates, a range of expected displacements from millimeters to meters, and targets at a distance of a few tens of meters to a few kilometers, depending on the type and dimensions of monitored structure, with measurements under the influence of atmospheric effects.  Can the proposed technique respond to such needs?  I think that the authors should first consult two papers on this topic, Xu & Brownjohn, Civ. Struct. Health Monit. 2018, 8, 91–110 and Fradelis et al, Sensors 2020, 20, 3217, and then try to place their idea in this context.

If the authors respond positively to the above question, a radical revision of the manuscript is required, and this may lead to a manuscript which I shall be happy to recommend for publication in Sensors.

In addition, careful copy editing is required (in references, etc.)

Response 1: Thank you very much for the comment. We are aware of the problem. As the reviewers have stated, a clear judgment of the intended content and target structure of the measurement is fundamental for a measurement to be viable. The focus of the manuscript is frame images which comes from surveillance video. This method for measuring and tracking structural displacement for distances up to a few meters is useful for structural monitoring, and the applicability of the method proposed in this paper is determined by the performance of the hardware (Line 103-104, it has been explained in the paper). And the relevant experimental conditions of this paper have been further explained in the modified paper. Obviously, the effects of environmental conditions on the photogrammetry exist, such as the effects of light conditions, raining or fog weather. In this paper, the study was carried out under the determination of the light conditions in the laboratory room, and the influence of rain and fog meteorological conditions were not involved during the study process. The light intensity of this study in the laboratory ranged from 50 to 20000 lux. The influence of atmospheric effects on the accuracy of the method under complex conditions such as foggy and rainy days will be the goal of our subsequent research, thank reviewer again for your meaningful good suggestions. In addition, the literature suggested by the reviewers has been carefully read and cited in the text. The modifications to the article are as follows:

Line 236:

“The vertical displacement changes at the beam midpoint were measured continuously and synchronously by using contact measurements. Obviously, the effects of environmental conditions on the photogrammetry exist, such as the effects of light conditions, raining or fog weather [47]. In this paper, the study was carried out under the determination of the light conditions in the laboratory room, and the influence of rain and fog meteorological conditions were not involved during the study process. The measured illumination in the laboratory of this study ranged from 50 to 20000 lux.”

Round 2

Reviewer 1 Report

Comments and Suggestions for Authors

This Reviewer is satisfied with the current state of the revised manuscript. 

Some minor (editorial) adjustments, such as the surnames of the Authors (which should not be all upper case), could be solved without the need for a second round of peer review.

Comments on the Quality of English Language

The English of the revised manuscript is fine

Author Response

Response 1: Thank you very much for the comment. Modifications have been made to the detailed issues raised by the reviewers, as follows:

The surnames of the authors:

Tongyuan Ni*1,2, Liuqi Wang1, Xufeng Yin1, Ziyang Cai1, Yang Yang1,2, Deyu Kong*1,2, Jintao Liu1,2

Line 493:

Author Contributions: Conceptualization, Tongyuan Ni. and Yang Yang; methodology, Tong-yuan Ni; software, Liuqi Wang; investigation, Liuqi Wang, Xufeng Yin and Ziyang Cai; writing—original draft preparation, Liuqi Wang and Xufeng Yin; writing—review and editing, Tongyuan Ni and Jintao Liu; supervision, Yang Yang and Deyu Kong; project administration, Tongyuan Ni and Deyu Kong and Yang Yang; funding acquisition, Tongyuan Ni, Yang Yang and Deyu Kong. All authors have read and agreed to the published version of the manuscript.

Reviewer 2 Report

Comments and Suggestions for Authors

Changes made are minor, and although they slightly result in improvement of the manuscript, the basic problems noticed in my first review remain.

Comments on the Quality of English Language

Acceptable

Author Response

Point 1: The authors suggest that they propose a method for structural monitoring based on analysis of common surveillance video images. In reality, they propose a method for calibration and sensitivity analysis of certain parameters of short-range positioning of a target from video analysis.  This is quite different. This means that the title and the focus of the manuscript should be changed.

Response 1: Thank you very much for the comment. The title of the manuscript has been modified as ‘Experimental study on measuring and tracking structural displacement based on surveillance video images analysis’.

And the focus of the manuscript has been modified introduction section as follows:

Line 72:

“So, a new kind of non-contact monitoring method for bridge structure deformation could be realized in the future: the localization information change of the prefabricated circular target in the surveillance video is used to characterize the displacement change of the structural location feature point through the surveillance video analysis and image processing techniques, so as to track and monitor the structural displacement [40].”

Line 467:

In this paper, the influence of different imaging distances and angles on the conversion coefficient (η) that converts the pixel coordinates to the actual movements was studied, respectively. Based on the surveillance video images analysis and digital image processing, a method to characterize the displacement of structural position feature points was designed. And a new non-contact measurement method of structural displacement would be proposed. The measurement results of surveillance video images were com-pared with a high-precision laser displacement meter.

Point 2: Perhaps the proposed technique designed/tested for distances up to a few meters might be useful for structural monitoring, but this requires understanding of what should be measured: 3-D to 1-D coordinates, a range of expected displacements from millimeters to meters, and targets at a distance of a few tens of meters to a few kilometers, depending on the type and dimensions of monitored structure, with measurements under the influence of atmospheric effects.  Can the proposed technique respond to such needs? I think that the authors should first consult two papers on this topic, Xu & Brownjohn, Civ. Struct. Health Monit. 2018, 8, 91–110 and Fradelis et al, Sensors 2020, 20, 3217, and then try to place their idea in this context.

If the authors respond positively to the above question, a radical revision of the manuscript is required, and this may lead to a manuscript which I shall be happy to recommend for publication in Sensors.

Response 2: Thank you very much for the comment. We are aware of the problem. As the reviewers have stated, a clear judgment of the intended content and target structure of the measurement is fundamental for a measurement to be viable. The focus of the manuscript is frame images which comes from surveillance video. This method for measuring and tracking structural displacement for distances up to a few meters is useful for structural monitoring, and the applicability of the method proposed in this paper is determined by the performance of the hardware (Line 103-104, it has been explained in the paper). And the relevant experimental conditions of this paper have been further explained in the modified paper. Obviously, the effects of environmental conditions on the photogrammetry exist, such as the effects of light conditions, raining or fog weather. In this paper, the study was carried out under the determination of the light conditions in the laboratory room, and the influence of rain and fog meteorological conditions were not involved during the study process. The light intensity of this study in the laboratory ranged from 50 to 20000 lux. The influence of atmospheric effects on the accuracy of the method under complex conditions such as foggy and rainy days will be the goal of our subsequent research, thank reviewer again for your meaningful good suggestions. In addition, the literature suggested by the reviewers has been carefully read and cited in the text. The modifications to the article are as follows:

Line 235:

“The vertical displacement changes at the beam midpoint were measured continuously and synchronously by using contact measurements. Obviously, the effects of environmental conditions on the photogrammetry exist, such as the effects of light conditions, raining or fog weather [47]. In this paper, the study was carried out under the determination of the light conditions in the laboratory room, and the influence of rain and fog meteorological conditions were not involved during the study process. The light intensity of this study in the laboratory ranged from 50 to 20000 lux.”

Line 456:

“The effects of environmental conditions (e.g., lighting conditions, the atmospheric refraction, and turbulence) on the method of this paper would be further explored in our continuous studies to improve the quality of vision-based measurements. Full and real-time non-contact monitoring of structural displacements will be realized in future practical applications.”

In addition, this paper further specifies the scope of application of the described methods and the objective structure of the measurements. The specific modifications are as follows:

Line 84:

“In this study, a novel non-contact monitoring method will be proposed and this method focuses on tracking and monitoring the development of structural displacement of small and medium-span bridges under loads during operation and maintenance so as to realize two-dimensional measurements of the displacement response of the structure. The bridge structure displacements above mentioned are small compared with the structural scale and the distance of the camera-to-structure.  In order to realize this new non-contact online monitoring method, the influence of different imaging distances and angles on the conversion coefficient (η) that converts the pixel coordinates to the actual displacement was first experimentally studied. And then, the measuring and tracking of bridge structural displacement based on surveillance video images was investigated by laboratory-scale experiments under idealized conditions. Lastly, the feasibility of measuring and tracking structural displacement based on surveillance video images would be experimentally confirmed in this paper.”

Point 3: In addition, careful copy editing is required (in references, etc.)

Response 3: Thank you very much for the comment. The format of the references has been further reviewed and revised.

Round 3

Reviewer 2 Report

Comments and Suggestions for Authors

The last version of the manuscript is improved, and although the authors have not responded to many of the points I noticed, I feel I can recommend its publication

Comments on the Quality of English Language

Needs careful copy editing